# Non-Destructive Luminescence-Based Screening Tool for *Listeria monocytogenes* Growth on Ham

**DOI:** 10.3390/foods9111700

**Published:** 2020-11-20

**Authors:** Shannon D. Rezac, Cristina Resendiz-Moctezuma, Dustin D. Boler, Matthew J. Stasiewicz, Michael J. Miller

**Affiliations:** Department of Food Science and Human Nutrition, University of Illinois at Urbana-Champaign, 1302 W Pennsylvania Ave, Urbana, IL 61801, USA; srezac02@gmail.com (S.D.R.); cr16@illinois.edu (C.R.-M.); dustin.boler@topigsnorsvin.us (D.D.B.); mstasie@illinois.edu (M.J.S.)

**Keywords:** *Listeria monocytogenes*, luminescence, ham, ready-to-eat meat products, antimicrobial

## Abstract

*Listeria monocytogenes* is a food-borne pathogen often associated with ready-to-eat (RTE) food products. Many antimicrobial compounds have been evaluated in RTE meats. However, the search for optimum antimicrobial treatments is ongoing. The present study developed a rapid, non-destructive preliminary screening tool for large-scale evaluation of antimicrobials utilizing a bioluminescent *L. monocytogenes* with a model meat system. Miniature hams were produced, surface treated with antimicrobials nisin (at 0–100 ppm) and potassium lactate sodium diacetate (at 0–3.5%) and inoculated with bioluminescent *L. monocytogenes*. A strong correlation (*r* = 0.91) was found between log scale relative light units (log RLU, ranging from 0.00 to 3.35) read directly from the ham surface and endpoint enumeration on selective agar (log colony forming units (CFU)/g, ranging from 4.7 to 8.3) when the hams were inoculated with 6 log CFU/g, treated with antimicrobials, and *L. monocytogenes* were allowed to grow over a 12 d refrigerated shelf life at 4 °C. Then, a threshold of 1 log RLU emitted from a ham surface was determined to separate antimicrobial treatments that allowed more than 2 log CFU/g growth of *L. monocytogenes* (from 6 log CFU/g inoculation to 8 log CFU/g after 12 d). The proposed threshold was utilized in a luminescent screening of antimicrobials with days-to-detect growth monitoring of luminescent *L. monocytogenes*. Significantly different (*p* < 0.05) plate counts were found in antimicrobial treated hams that had reached a 1 log RLU increase (8.1–8.5 log(CFU/g)) and the hams that did not reach the proposed light threshold (5.3–7.5 log(CFU/g)). This confirms the potential use of the proposed light threshold as a qualitative tool to screen antimicrobials with less than or greater than a 2 log CFU/g increase. This screening tool can be used to prioritize novel antimicrobials targeting *L. monocytogenes*, alone or in combination, for future validation.

## 1. Introduction

*Listeria monocytogenes* is a gram-positive microorganism, found ubiquitously in the environment, which causes the foodborne disease listeriosis. In 2019, the Center for Disease Control and Prevention (CDC) reported 134 cases of listeriosis in the U.S with one of the highest case-fatality rates among food-borne diseases–16% [1]. Ready-to-eat meat products support the growth of *L. monocytogenes* and have been classified as high-risk food products since these products are commonly consumed without any further treatment [2]. Listeriosis outbreaks have been linked to ready-to-eat (RTE) products, such as fresh cheeses, luncheon meats, and fresh produce [3].

To control the growth of *L. monocytogenes*, RTE meat products are often formulated with antimicrobial agents [4]. While there are many categories of commercially-available antimicrobials, novel antimicrobials are continuously being developed to meet evolving consumer demands while continuing to be effective and affordable [5,6].

The classic method to evaluate antimicrobials in deli meats (the “ham slice method”) is laborious [7,8,9,10,11]. This process uses a large amount of food product, such as many slices of ham, and utilizes the traditional method of enumeration on selective media. Surface application of antimicrobials on whole deli meat slices is done at various levels and combinations. Slices are inoculated with *L. monocytogenes*, or the non-pathogenic surrogate *Listeria innocua*, and then stored at refrigeration temperature. At selected time points, the meat slice is physically disrupted for cell recovery, diluted with buffer, and the diluted solution is plated onto selective agar. After 24–48 h of growth, the plate counts can be compared. This traditional enumeration method is destructive to the food product and requires days to receive results of the antimicrobial effectiveness. This method is not suitable for large-scale screening of antimicrobials, alone or in combination, in ham. Consequently, when a large number of antimicrobials need to be screened, the optical density can be measured for small volumes of broth with antimicrobials and inoculum after a ~24 h incubation period [12]. Problematically, studies have highlighted the differences in *L. monocytogenes* sensitivity to antimicrobials in various broths and food matrices [13]. Reasons for the differences include the presence of competing microbial flora, impact of the food composition on antimicrobial function, temperature of the assay and changes in the *L. monocytogenes* biology such as growing planktonically in broth whereas forming biofilms on food matrices [13,14]. Screening followed by validation in food is becoming more prevalent as synergistic relationships of antimicrobial combinations or antimicrobials with post-lethality treatments are being targeted to lower the cost of the treatment while increasing effectiveness [7,15].

Researchers have proposed optimized laboratory models of food products with a high-risk of *L. monocytogenes* contamination, such as queso fresco [16]. These methods can be used to screen many antimicrobials, including combinations, while avoiding reliance on a broth system. There have been some ham-like models proposed in replacement of using whole ham slices [17,18,19], however, none have been able to replicate the ham slice method. A model is needed to mimic the ham slice method while increasing efficiency of antimicrobial screening.

A long-proposed mechanism to measure *L. monocytogenes* growth in food is a luciferase-based reporter system [20,21]. Luciferases are enzymes that catalyze a reaction that produces visible light in a process called bioluminescence. The *lux* genes responsible for bioluminescence have been transferred and expressed in several microorganisms [22]. The luciferase reporter system has been modified for gram-positive microorganisms and integrated into the chromosome of *L. monocytogenes* to measure growth in a food matrix [23]. In that work, the relative light units (RLU) were correlated to the traditional plate count colony forming units (CFU), to allow real-time analysis of *L. monocytogenes* growth. However, the food matrix was destroyed to measure the light-emission from the cells. This process has not been modified to screen antimicrobials targeting *L. monocytogenes* by reading light directly from a food surface while not destroying the food matrix.

This study evaluates the use of a non-destructive luminescence screening tool for *L. monocytogenes* growth in a miniature ham system. First, the method of measuring light emissions from the surface of miniature ham disks in 96-well microplates was developed. The correlation of RLUs and CFUs was found for multiple inoculation levels to determine the strongest correlation. Then endpoint RLUs and CFUs were measured with limited antimicrobial treatments of known effectiveness (potassium lactate sodium diacetate (PLSDA) at 1.75% and 3.50%, respectively). Finally, this method was validated by screening two antimicrobials with known activity at three levels (Nisin (NI) at 100, 50, and 25 ppm and PLSDA at 2.50%, 1.25%, and 0.63%) in a ham model to determine which antimicrobial treatments allowed less than 2 log growth over 21 days. This high-throughput screening method can be utilized to screen many novel antimicrobials or antimicrobial combinations and thus identify promising treatments for further traditional validation with eventual implementation in deli meats.

## 2. Materials and Methods

### 2.1. Sample Preparation

Ham was made at the University of Illinois Meat Science Laboratory, a commercial sales facility regulated by the United States Department of Agriculture’s (USDA) Food Safety and Inspection Service (FSIS). The ham formula was based on the retail ham sold by the Meat Science Laboratory. Sodium hexametaphosphate (SHMP) (29 g) was added to a mixture of half ice and water (1.07 kg) until fully dissolved. Then NaCl salt (160 g), Prague powder No. 1 curing salt (13 g), and sodium erythorbate (5 g) were mixed into the brine. Dextrose (146 g) was added last to the brine. Lean pork leg inside cuts (4.53 kg) were diced into 1 inch cubes and vacuum tumbled with the brine for two hours. The mixture was stuffed into nylon casings and smoked at 170 °C heat for 7 h. After chilling, the ham was sliced into 2.5 mm slices, individually vacuum sealed and frozen. The ham had a protein fat-free (PFF) of 22.3, which is the protein content of the nonfat portion of the finished ham and reflects the amount of added water. This determines the type standard identity of the product, with this product being labeled as “ham”. This product had a final sodium content of 1.3% and a water activity (aw) of 0.963 ± 0.001 at 4 °C.

### 2.2. Microorganisms and Growth Conditions

*Listeria monocytogenes* Xen19 was acquired from PerkinElmer (PerkinElmer, Waltham, MA, USA). This strain was derived from the American Type Culture Collection (ATCC) 23074, with a copy of the *Photorhabdus luminescens lux* operon in its chromosome, making it stable and constitutively expressed. This strain was inoculated in a brain heart infusion (BHI) (Difco, Becton Dickinson and Co., Sparks, MD, USA) from glycerol stocks stored at −80 °C. The bacterium was grown overnight in 10 mL BHI broth for 18 h at 37 °C with an agitation of 250 rpm. The cultures were then passed to 5 mL BHI media to grow for 24 h at room temperature (22 °C) without agitation, as adapted from Liu et al. [24]. The *L. monocytogenes* Xen19 overnight culture (~9 log CFU/mL) was then diluted in phosphate buffered saline solution (PBS) (KCl 200 mg/L; KH2PO4, 200 mg/L; NaCl, 8 g/L; Na2HPO4, 1.15 g/L, pH 7.2, MP Biomedicals, LLC, OH) to achieve the desired inoculation level, ranging from 2 to 8 log CFU/g, specified in each experiment. These same conditions were used for a cocktail of *L. monocytogenes* strains associated with foodborne disease outbreaks (*L. monocytogenes* NRRL B-33419, *L. monocytogenes* NRRL B-33424, *L. monocytogenes* NRRL B-33420, *L. monocytogenes* NRRL B-33513, and *L. monocytogenes* NRRL B-33104). Equal parts of the overnight cultures of the cocktail strains were mixed before diluting in PBS. The strains used in this study are listed in Table 1.

### 2.3. Description of Ham Model

Vacuum sealed 2.5 mm ham slices were defrosted at 5 °C for at least 12 h before using immediately. A flame-sterilized 4.8 mm diameter circular die was used to punch uniform ham disks into microtiter plates. White microtiter plates were used (96-well, white, clear-bottomed, Greiner bio-one, Monroe, NC, USA) to analyze the bioluminescent *L. monocytogenes* and the cocktail strains. Each ham disc weighed 88 ± 4 mg (Figure 1). Antimicrobials were diluted in sterile distilled water and 30 µL was pipetted onto the ham disk. The plate was centrifuged at 3220× *g* for 4 min at 4 °C (Centrifuge 5810 R, Eppendorf, Hauppauge, NY, USA) to ensure all droplets of the antimicrobial were covering the ham disk. Hams were inoculated with 30 μL of the diluted *L. monocytogenes* before centrifuging at 3220× *g* for 6 min at 4 °C. Centrifugation was done to pellet the cells to the ham surface. Excess liquid was removed from each well, with over half of the liquid recovered. This process was validated for the initial inoculation amount to match the Day 0 ham measurement with enumeration by selective media. A plastic lid was placed on the 96-well plate and sealed in an air-tight container also containing a saturated salt solution of KNO3 (chosen for its similar water activity of 0.96 ± 0.01 at 10 °C) to prevent the ham from drying with exposure to the environment, while taking measurements over the time course. The container was stored at 5 °C for the number of days noted for each experiment.

### 2.4. Antimicrobials Used

Two commercially-available antimicrobials with GRAS status were used in this study. Nisaplin (Danisco, Beaminster, UK) contains the bacteriocin nisin (NI) (2.5% *w*/*w*) and was diluted in sterile distilled water. It has GRAS status for up to 220 parts per million (ppm) by FSIS on the surface of cooked RTE meat products (GRAS No. 000065). The organic acid combination, L-potassium lactate (54.5–57.5% *w*/*w*) and sodium diacetate (3.7–4.3% *w*/*w*) (PLSDA) (Opti.Form^®^ PD 4, Corbion Purac, Amsterdam, The Netherlands) was used up to an upper limit of 3.5% *w*/*w* (company recommendation for sliced ham). Both were tested by surface application in the model ham system.

### 2.5. Determination of Growth at the End Point

A 12-day endpoint was determined based on U.S. government recommendations for safe deli meat storage. The Food and Drug Administration (FDA) Food Code 2013 suggests that opened RTE food products, including deli meats, in a food establishment stored at 5 °C or less be held for a maximum of 7 days. Additionally, the United States Department of Agriculture (USDA) Refrigeration and Food Safety (2015) recommends that a consumer should keep fully-cooked deli ham slices for no more than 3–5 days in their refrigerator before discarding. Using the greatest amount of time for each federal recommendation, a time course of 12 days was implemented for thawed and oxygen-exposed ham slices.

Luminescence readings were taken with a luminometer (FilterMax F5 Multi-Mode Microplate Reader, Molecular Devices, San Jose, CA, USA) that detected all light in the visible spectrum between 400 and 750 nm from the ham surface. Row optimization, an integration time of 2000 ms, and a read height of 1 mm were set in the Softmax Pro 7 software (Molecular Devices, San Jose, CA, USA). Wells with uninoculated ham was set as the blanks in the software.

At some time points, hams were destructively sampled with traditional enumeration, after reading the luminescence from the ham, to allow for comparison between RLU and CFU (Figure 1). This was done by aseptically removing the miniature hams from the 96-well plate and suspending 1:10 in PBS in a 1.5 mL Eppendorf tube. The miniature hams were disrupted by vortexing (Vortex-Genie 2, VWR, Rochester, NY, USA) for 5 min to remove the cells from the ham surface, giving recovery that was not statistically different from a stomacher. Additional serial dilutions in PBS were made to the ham solutions. Then 50 µL of the appropriate dilutions were enumerated by plating on Polymyxin Acriflavin Lithium-chloride Ceftazidime Esculin Mannitol (PALCAM) Listeria-Selective agar (EMD-Millipore) with the Listeria selective supplement (EMD-Millipore) in duplicate with a spiral plater (Eddy Jet-Spiral plater, Neutec group inc., Farmingdale, NY, USA) and incubated at 37 °C for 48 h prior to counting with a colony counter (The IUL Flash and Go, Neutec group inc., Farmingdale, NY, USA).

The growth after 12 days on ham of the bioluminescent strain was compared to a cocktail of *L. monocytogenes* strains associated with foodborne disease outbreaks with enumeration by plating (Figure 1). The initial inoculum was 4.8 ± 0.2 log CFU/g for the cocktail strains and *L. monocytogenes* Xen19.

For determining the preliminary luminescent RLU threshold using the miniature ham model, a control ham, and hams with 1.75% and 3.5% PLSDA were measured with a luminometer and by traditional plate counts at days 0, 6, and 12. This experiment was done in triplicate. The starting inoculation was 6.0 ± 0.2 log CFU/g.

For the application of the model for antimicrobial screening, the hams were tested with six technical replicates and the experiment was done in biological duplication, with a starting inoculation of 5.9 ± 0.2 log CFU/g. The RLU was read once a day for 21 days, encompassing the 12-day anticipated shelf life plus additional time for an outgrowth signal to develop in the growth-inhibited treatments (Figure 1). When the average light signal reached 1 log RLU, three of the replicates were aseptically removed and enumerated by serial dilutions and plating on selective medium as described above. The remaining three hams continued to have light measurements taken until Day 21.

### 2.6. Data Analysis

JMP Pro 13 statistical software (SAS Institute, Cary, NC, USA) was used for all data analysis. An ANOVA analysis was performed to determine statistical differences (*p* < 0.05) between pathogen growth on treated and untreated hams (treatment effect) inoculated with an *L. monocytogenes* cocktail or *L. monocytogenes* Xen19 (strain effect), and their interaction. This experiment was performed in triplicate.

The Pearson correlation coefficient was calculated to evaluate the inoculum level that had the highest linear correlation between log (CFU) and log (RLU) in ham over a 12-day time course. The experiment was performed in technical duplicates and biological triplicates.

An ANOVA analysis was done for the days-to-detect experiment to compare plate counts between all treatments, with the end goal of comparing the treatments that reached the 1 log RLU increase threshold (control, PLSDA 0.6%, PLSDA 1.25%, and NI 25 ppm) with the treatments that did not reach the 1 log RLU increase threshold at 21 days (PLSDA 2.5%, NI 50 ppm, and NI 100 ppm). Once the ANOVA showed a significant treatment effect, an unbiased all-pair comparison was performed using Tukey-Kramer test with a significance level of *α* = 0.05. After this analysis, the treatments that exceeded 1 log RLU were grouped together. The hams were tested with six technical replicates and the experiment was done in biological duplication.

## 3. Results

### 3.1. Growth Comparison of the Bioluminescent Strain to Foodborne Disease Associated Strains

The growth of *L. monocytogenes* Xen19 was compared to a 5-strain cocktail of *L. monocytogenes* strains after 12 days using enumeration by plating with no antimicrobial treatment and PLSDA at 3.50% (Figure 2). The change in growth was examined and analyzed with the 12-day end-point plate count (Ni) being subtracted from the starting count at Day 0 (N_0_). An ANOVA indicated significant statistical differences between hams treated with 3.5% PLSDA and the non-treated ones (*p* < 0.05), which was to be expected. However, no statistical differences (*p* = 0.98) were found between hams inoculated with the *L. monocytogenes* cocktail and the ones inoculated with *L. monocytogenes* Xen19, nor was the interaction effect significant (*p* = 0.40).

### 3.2. Correlation of Colony Forming Units and Relative Light Units

To determine the best correlation between CFUs and RLUs, both parameters were measured and plotted against each other. For this purpose, multiple *L. monocytogenes* inoculation levels were tested in triplicate over a 12-day time course (Figure 3). Figure 3A shows the correlation between CFU (log) and RLU (log) with limited antimicrobial treatments (Control Day 0, Control Day 12, PLSDA 1.75% Day 12, and PLSDA 3.50% Day 12) with inoculations ranging from 2 to 8 log CFU/g. The inoculation levels with the best initial correlation coefficients (5 and 6 log CFU/g) were then brought forward into more trials. Figure 3B shows the consequent prediction models of 5 and 6 log CFU/g inoculations for RLUs and CFUs in trials using controls, PLSDA at 1.75% and 3.50%, and NI at 110 and 220 ppm. The 6 log CFU/g inoculation level gave the best correlation of CFUs to RLUs (*r* = 0.91). Therefore, the variance is well explained by the model. A 6 log CFU/g inoculation level was brought forward into the rest of the experiments.

### 3.3. Preliminary Relative Light Unit Threshold Determination with Limited Antimicrobial Treatments

To determine if measuring light emission of *L. monocytogenes* Xen19 could produce a preliminary RLU threshold to separate treatments that exhibited growth or not, a control ham and hams treated with either 1.75% or 3.50% PLSDA were measured at days 0, 6, and 12 (Figure 4). The RLU trend in samples with or without treatments followed what was expected with the control ham emitting the most light, followed by the lower level of PLSDA, and the higher level of PLSDA emitting very little light. This was the same trend as the traditional plate counts. Whereas only the plate counts had significant differences with and without treatments at Day 6, both plate counts and RLU had significant treatment differences at Day 12 (*p* < 0.05).

The line in the RLU graph at 1 log RLU was proposed to distinguish between the samples that showed minimal growth at best and samples that notably grew. In this experiment, the 1 log RLU threshold would accurately classify the control ham as showing over a 2 log CFU/g increase after 12 days, and the ham with PLSDA at 1.75% and 3.5% as having less than a 2 log CFU/g increase. This threshold was brought forward into further experiments.

### 3.4. Application of the Model to Antimicrobial Screening

To test if this model could be applied to a higher-throughput screening approach with antimicrobials targeting *L. monocytogenes* using the proposed threshold of 1 log RLU, many levels of antimicrobials (NI at 100, 50, and 25 ppm, and PLSDA at 2.50, 1.25, and 0.63%) were surface applied to the hams in 96-well plates and light emission was tested daily for 21 days. When the average RLU of the sample was greater than 1 log, it was enumerated by plating. At the end of 21 days, the rest of the samples that did not reach 1 log RLU were enumerated by plating, and these results are shown in Table 2.

The control was the first to reach the threshold at 10 days, followed by PLSDA at 0.6% at 11 days, PLSDA 1.25% at 15 days, and NI 25 ppm at 17 days (Table 2). An ANOVA test was used to confirm that the treatments that reached 1 log RLU threshold (control, PLSDA 0.6%, PLSDA 1.25%, and NI 25 ppm) were significantly different (*p* < 0.05) from the treatments that did not reach that threshold (2.5% PLSDA, 50 ppm NI, and 100 ppm NI). Post-hoc analysis using the Tukey-Kramer test determined that the treatments that did not reach the 1 log RLU threshold had significantly different mean plate counts than each other and to the treatments that exceeded the 1 log RLU threshold (*p* < 0.05). In addition, the mean plate counts at the time point where the treatments first exceeded 1 log RLU were not significantly different (*p* > 0.05).

The samples that reached an average of 1 log RLU, showed over a 2 log CFU/g increase (>8 log CFU/g). The samples that did not reach 1 log RLU over 21 days were the hams with PLSDA at 2.50%, and NI at 50 and 100 ppm. In addition, their corresponding plate counts were below 8 log CFU/g. Figure 5 shows the RLU measurements over 21 days for each sample. The black horizontal line at 1 log RLU represents the threshold when the samples were enumerated by plating. The threshold accurately classified the samples that grew beyond 2 log CFU/g and the samples that did not.

## 4. Discussion

In this study, a luminescence-based screening was developed for rapid insight into the growth of *L. monocytogenes* in deli-meat. The correlation of plate counts and luminescence was determined to be the strongest (*r* = 0.91) at a 6 log CFU/g inoculation, an inoculation level that has been previously used for microbial testing [25] and is high enough to be able to read from a porous ham surface. While this inoculation level is admittedly high, our results with NI and PLSDA match the literature in that 50 and 100 ppm NI and 2.5% PLSDA were able to limit *L. monocytogenes* growth on ham for 21 days [26,27,28]. A light emission threshold was proposed for antimicrobial screening to indicate antimicrobial treatments that allowed a significant increase of *L. monocytogenes* and would not need further analysis. The model was then utilized to screen antimicrobials at various levels by measuring light-emission daily to determine when the antimicrobial treatment allowed *Listeria* growth to pass the proposed threshold. It was determined that at 1 log RLU, the sample had increased over 2 log CFU, indicating this model can be a tool to screen antimicrobials with less than or greater than a 2 log CFU/g increase. The treatments that do not hit the RLU threshold could be brought forward to traditional testing methods, such as the ham slice method, making this model a rapid preliminary test for screening a large number of antimicrobial treatments.

The proposed model of miniature hams and a luminescence assay optimizes the use of both the food product and antimicrobials. The model allows for many more hams to be screened than the ham slice method while avoiding using a broth system, which does not mimic the food matrix [14]. This combines the high-throughput nature of a broth system while replicating a whole ham slice. Hams with varying formulas including natural ingredients (such as celery powder and cherry powder), higher water content (products labeled “ham with natural juices” or “ham with water added”), or reduced-sodium ham can be easily compared between hundreds of antimicrobial treatments. This method could also be potentially applied to other solid food products. The non-destructive nature of the luminescent assay allows repeated measurements to be taken, which is useful when working with novel compounds that require labor-intensive synthesis [29]. It was found that reading the luminescence off the top of an evenly surface-coated ham disk allows for instant qualitative insight on antimicrobial effectiveness, whereas traditional enumeration techniques require an additional 2 days for enumeration.

Comparing luminescence between antimicrobial treatments gave similar trends as for the traditional enumeration method of selective plating. Selective plating showed significant differences between the control hams and hams with antimicrobial treatments on Day 6 while the luminescence reading did not until Day 12, indicating that the plate counts can significantly measure smaller cell number changes than what is possible for luminescence with the current assay. This is in accordance with the mechanism of luminescence, which relies on metabolic activity and will produce the most light when the bacteria are in log phase, rather than lag phase or stationary phase [30]. This method is presumed to compare between treatments that have allowed *Listeria* to enter log phase and those that are maintained in lag phase.

The application of the model to high throughput antimicrobial screening showed a threshold in light emission could be set that would represent an accurate cell number increase of *L. monocytogenes*. On the respective days that each treatment reached an average of 1 log RLU, there was over a 2 log CFU/g increase in cell population. Threshold values have been previously established to screen irradiated food using photo-stimulated luminescence [31,32]. Bioluminescent *Listeria* has been utilized for high-throughput competition assays in milk [33], while other studies have utilized luminescent *Salmonella* for detection assays in homogenized chicken meat [34]. Riedel et al., 2007 [23] utilized luminescent *Listeria* in a food matrix, however, the food products were destroyed for light-emission readings and this was not utilized for antimicrobial screening. According to the authors’ knowledge, this is the first time the luciferase system has been applied to rapidly screen antimicrobials from a food matrix in a non-destructive manner. While other proposed reporter systems have used fluorescent proteins, such as green fluorescent protein (GFP) [35], a benefit of using the luciferase-based reporter system rather than a fluorescent protein is that it avoids naturally occurring fluorescence that can lead to high background levels during in vitro and in vivo fluorescence measurements [36].

While *L. monocytogenes* Xen19 has been validated to exhibit growth trends similar to a cocktail of foodborne disease outbreak associated strains in our study, use of this method would not give a full insight into how antimicrobials will affect the growth of all *Listeria* strains. However, since this model uses a commercially-available bioluminescent strain, the process can be quickly replicated as *L. monocytogenes* Xen19 can be easily purchased and requires no further genetic engineering. This luminescence assay should be a preliminary screening tool that avoids using a broth system, where promising antimicrobial treatments are then further tested with traditional methods, as the model also cannot distinguish between bacteriostatic and bactericidal antimicrobial treatments. The greatest advantage of this assay is its extreme high-throughput nature, although this also limits the sensitivity. To potentially increase the sensitivity of the assay or lower the threshold of light-emission to indicate growth, this assay could be adapted to be destructive to the hams, such as removing the cells from the food surface in a buffer and reading the luminescence from the buffer as have been done previously [23]. This adaptation, however, would increase the assay time and lower the throughput, which was not the focus of this study.

## 5. Conclusions

A rapid, high-throughput and non-destructive assay was developed for insight about the growth of *L. monocytogenes* in deli-meat. A novel technique was proposed with miniature ham disks in 96-well plates and light-emission read from the food surface. It was determined that at 1 log RLU, the sample had increased over 2 log CFU, indicating this model can be a qualitative tool to screen antimicrobials with less than or greater than a 2 log CFU/g increase. This method can be adapted for novel antimicrobial screening for deli meat products or other solid food surface products to determine promising possibilities that should be brought forward for further testing with traditional methods.

## Figures and Tables

**Figure 1 foods-09-01700-f001:**
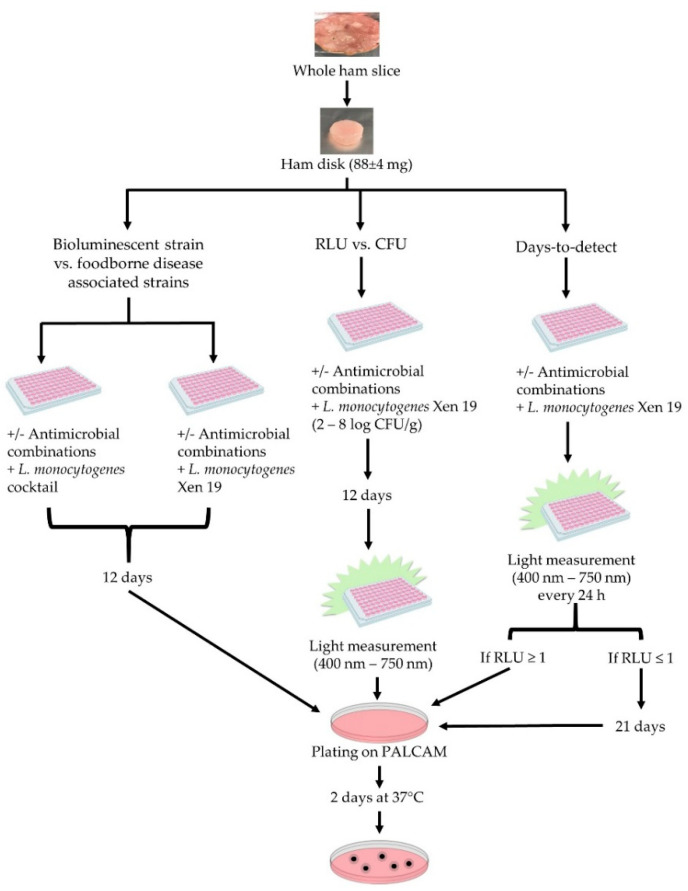
Schematic diagram of the experimental design to compare the growth of *L. monocytogenes* Xen19 to a cocktail of 5 *L. monocytogenes* strains that have been previously related to foodborne outbreaks; the experiment is to find the best correlation between relative light units (RLU) and colony forming units (CFU) for the days-to-detect experiment.

**Figure 2 foods-09-01700-f002:**
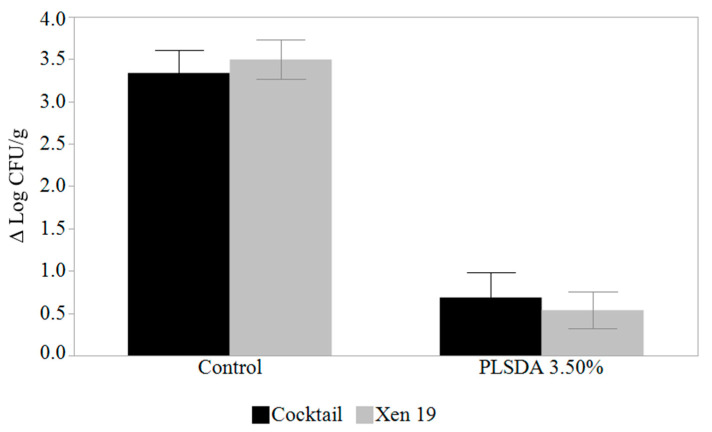
Comparison of growth after 12 days (N_i_–N_0_) at 5 °C with a cocktail of *L. monocytogenes* associated with foodborne disease outbreaks and *L. monocytogenes* Xen19 on a miniature ham model. The initial inoculum was 4.8 ± 0.2 log CFU/g. ANOVA analysis found no significant difference between the cocktail of *L. monocytogenes* and *L. monocytogenes* Xen19 controlling for treatment effect (*p* = 0.98).

**Figure 3 foods-09-01700-f003:**
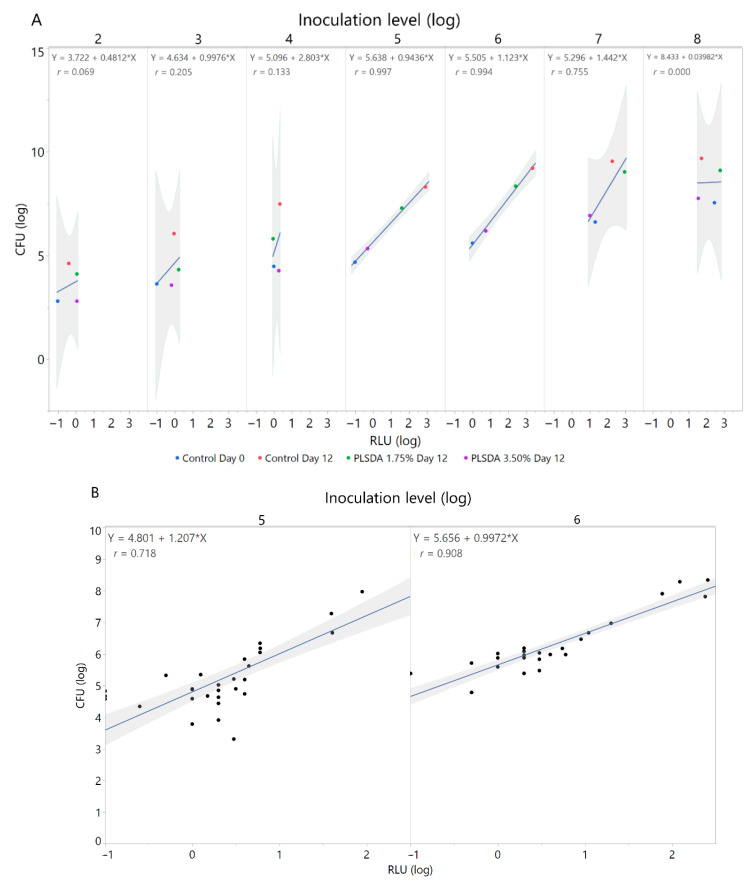
Correlation of CFU and RLU at multiple inoculation levels. The shaded region shows the 95% confidence intervals. (**A**) Preliminary screening with limited antimicrobials with inoculation levels ranging from 2 to 8 log CFU/g. (**B**) Additional trials for the inoculation levels with the best initial correlation coefficients (5 and 6 log CFU/g). See text for treatment details.

**Figure 4 foods-09-01700-f004:**
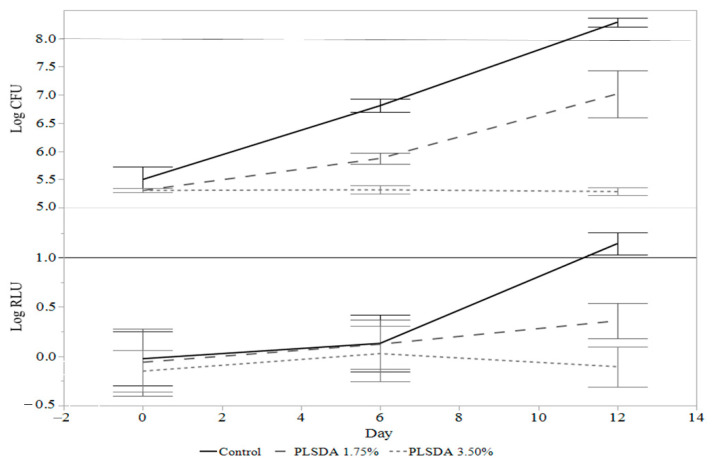
Comparison of *L. monocytogenes* Xen19 growth curves measuring colony forming units and relative light units in a miniature ham model with a 6 log colony forming unit (CFU)/g inoculation. Potassium lactate sodium diacetate (PLSDA) was used at 1.75% and 3.50% and compared to a ham with no antimicrobial treatment. The line at 1 log relative light unit (RLU) is proposed to distinguish samples with *L. monocytogenes* growth above 2 log CFU/g. The error bars are the standard deviation of the samples.

**Figure 5 foods-09-01700-f005:**
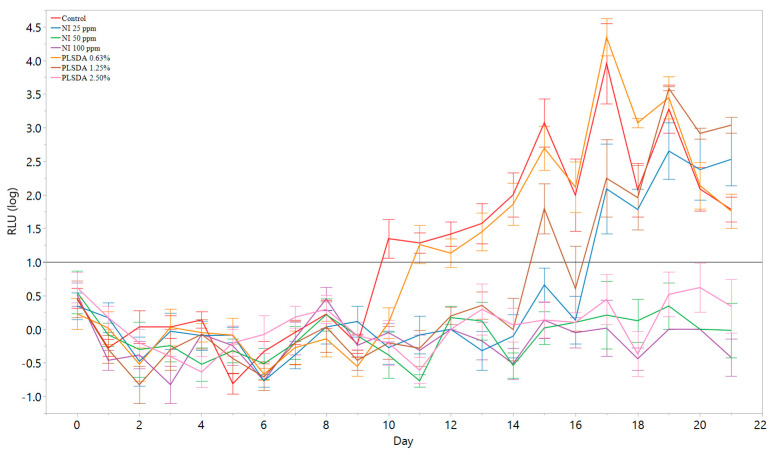
Using relative light units to distinguish growth/limited growth at a 1 log RLU threshold with multiple antimicrobials targeting *L. monocytogenes* in a miniature ham model. Plate counts at the first day with growth or at the end of 21 days are shown in Table 2.

**Table 1 foods-09-01700-t001:** *Listeria monocytogenes* strains used in this study.

Strain	Serotype	Phenotype/Source of Isolation
*L. monocytogenes* Xen19	4b	Bioluminescent; Derived from ATCC 23074
*L. monocytogenes* NRRL B-33419 *	1/2a	Human, epidemic, sliced turkey
*L. monocytogenes* NRRL B-33424 *	1/2b	Human, epidemic, chocolate milk
*L. monocytogenes* NRRL B-33420 *	4b	Food, epidemic, RTE meat products
*L. monocytogenes* NRRL B-33513 *	4b	Food, epidemic, pate
*L. monocytogenes* NRRL B-33104 *	4b	Food, epidemic, Jalisco cheese

* The foodborne disease outbreak associated strains of *L. monocytogenes* used in the cocktail.

**Table 2 foods-09-01700-t002:** The number of days to reach the threshold of detection for antimicrobial treatments targeting *L. monocytogenes* Xen19 *.

RLU Threshold Status	Antimicrobial Treatment	Plating Days	Average Plate Counts (log CFU/g)
Reached 1 log RLU	Control	10	8.49 ± 0.18 ^a^**
	PLSDA 0.6%	11	8.37 ± 0.20 ^a^
	PLSDA 1.25%	15	8.22 ± 0.29 ^a^
	NI 25 ppm	17	8.13 ± 0.23 ^a^
Did not reach 1 log RLU	PLSDA 2.50%	21	7.50 ± 0.39 ^b^
	NI 50 ppm	21	6.56 ± 0.48 ^c^
	NI 100 ppm	21	5.28 ± 0.24 ^d^

* Initial inoculum of 5.9 ± 0.2 log CFU/g. ** Means were analyzed by a Tukey-Kramer test. Means not connected by the same letter are significantly different (*p* < 0.05) and are indicated by different letter superscripts.

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
