# Peer review of "Non-Destructive Luminescence-Based Screening Tool for Listeria monocytogenes Growth on Ham"

_foods, 2020, doi:10.3390/foods9111700_

Round 1

Reviewer 1 Report

This is an rather interesting article and has potential to contribute an update of knowledge in the field. The content of this article is fully consistent with its title. Presented a problem developed on the basis of current knowledge. Using appropriate analytical methods, but lack of adequate statistical analysis. The scope of the analysis and interpretation of test results is correct, but not impressive. The subject of this article is important. Used the proper scope of the literature.

The idea of this manuscript is good and well written but there are some negative points such as:

  • Abstract – add specifically what came out. This is better described in the Conclusion subsection.
  • Data Analysis – poorly described and performed statistical analysis. The number of replications of the experiments and the tests used should be specified. Were parametric and non-parametric tests performed in addition to ANOVA?
  • Figures – drawings are not legible enough, they can introduce colors instead of differentiating with lines. Colored lines (especially in Figure 4) will make it easier to read.
  • Conclusions – you should write the conclusions, not what you did, it looks more like Abstract than Conclusion.

Author Response

Thanks for the nice comments and useful feedback. We have addressed every comment separately. See text for specifics.

  • Abstract – add specifically what came out. This is better described in the Conclusion subsection.

AU: We agree. The abstract has been substantially edited. Please see lines 13-28.

  • Data Analysis – poorly described and performed statistical analysis. The number of replications of the experiments and the tests used should be specified. Were parametric and non-parametric tests performed in addition to ANOVA?

AU: Agreed. Details about the number of replications and tests have been added to each section.  See lines 193  208 for these changes. Only parametric tests were performed. The tests performed were: ANOVA, Tukey-Kramer test, and Pearson correlation.

  • Figures – drawings are not legible enough, they can introduce colors instead of differentiating with lines. Colored lines (especially in Figure 4) will make it easier to read.

AU: Great suggestion. We have changed the color in Figure 2 and in Figure 4 to facilitate differentiation between treatments.

  • Conclusions – you should write the conclusions, not what you did it looks more like Abstract than Conclusion.

AU: We agree. The conclusion has been revised and edited. Please see lines 356 - 363.

Reviewer 2 Report

This study describes an innovative high-throughthput non-destructive assay for the screening of antimicrobial treatment for the control of L. monocytogenes in ham. This paper describes the development of an assay for the screening of antimicrobial for the control of L. monocytogenes by means of a luminescent L. monocytogenes strain. The topic is interesting because sets the ground for the development of high throughtput assays with all the advantages of an in vitro methodology but mimicking "real" testing on food. Even though assays based on luminescent strains have already been developed, the originality of this work lies in the implementation of the experiments on mini-hams, thus allowing to account for the effect of the matrix on L. monocytogenes growth. The paper is well written, although a bit more clarity in the material and methods section could be of use to help the reader's understanding of the whole process. The conclusions are consistent with the evidence found and the study addresses all the aims declared, highlighting strenghts and weakenesses of the methodology. 

My only comment is about the materials and methods section. Indeed, the experimental design is sound, but complicated. This complexity is reflected in the text, and is often difficult to grasp immediately all the steps of the process. I suggest that the authors could provide schematic representation of the different phases of the study development to ease the reader's comprehension.

Author Response

REVIEWER 2

My only comment is about the materials and methods section. Indeed, the experimental design is sound, but complicated. This complexity is reflected in the text, and is often difficult to grasp immediately all the steps of the process. I suggest that the authors could provide schematic representation of the different phases of the study development to ease the reader's comprehension.

AU: Thanks for the nice comments and helpful feedback. We agree and we have included a schematic diagram for the experimental design of the main experiments described in the manuscript. Please see Figure 1.